# Predictive Model for Mortality in Severe COVID-19 Patients across the Six Pandemic Waves

**DOI:** 10.3390/v15112184

**Published:** 2023-10-30

**Authors:** Nazaret Casillas, Antonio Ramón, Ana María Torres, Pilar Blasco, Jorge Mateo

**Affiliations:** 1Department of Internal Medicine, Hospital Virgen De La Luz, 16002 Cuenca, Spain; 2Medical Analysis Expert Group, Institute of Technology, University of Castilla-La Mancha, 16002 Cuenca, Spain; 3Department of Pharmacy, General University Hospital, 46014 Valencia, Spain; 4Instituto de Investigación Sanitaria de Castilla-La Mancha (IDISCAM), 45071 Toledo, Spain

**Keywords:** COVID-19, coagulation disorder, cytokine release syndrome, machine learning, SARS-CoV-2, XGB

## Abstract

The impact of SARS-CoV-2 infection remains substantial on a global scale, despite widespread vaccination efforts, early therapeutic interventions, and an enhanced understanding of the disease’s underlying mechanisms. At the same time, a significant number of patients continue to develop severe COVID-19, necessitating admission to intensive care units (ICUs). This study aimed to provide evidence concerning the most influential predictors of mortality among critically ill patients with severe COVID-19, employing machine learning (ML) techniques. To accomplish this, we conducted a retrospective multicenter investigation involving 684 patients with severe COVID-19, spanning from 1 June 2020 to 31 March 2023, wherein we scrutinized sociodemographic, clinical, and analytical data. These data were extracted from electronic health records. Out of the six supervised ML methods scrutinized, the extreme gradient boosting (XGB) method exhibited the highest balanced accuracy at 96.61%. The variables that exerted the greatest influence on mortality prediction encompassed ferritin, fibrinogen, D-dimer, platelet count, C-reactive protein (CRP), prothrombin time (PT), invasive mechanical ventilation (IMV), PaFi (PaO_2_/FiO_2_), lactate dehydrogenase (LDH), lymphocyte levels, activated partial thromboplastin time (aPTT), body mass index (BMI), creatinine, and age. These findings underscore XGB as a robust candidate for accurately classifying patients with COVID-19.

## 1. Introduction

Following the devastating epidemics of severe acute respiratory syndrome (SARS) in 2002 and Middle East respiratory syndrome (MERS) in 2012, both of which caused lethal diseases associated with coronaviruses, a novel coronavirus, severe acute respiratory syndrome coronavirus 2 (SARS-CoV-2) made its appearance in late December 2019. The disease was named coronavirus disease 2019 (COVID-19) and first emerged in Wuhan, China, where genetic sequencing of the virus was conducted, revealing it to be a novel beta-coronavirus [1,2].

On 11 March 2020, the World Health Organization (WHO) declared a pandemic due to the spread of this new disease, the severity of which compromised healthcare services worldwide.

COVID-19 is an extremely contagious disease with significant risks of morbidity and mortality. It can manifest as bilateral pneumonia, severe respiratory failure requiring mechanical ventilation, and/or multiorgan damage that can tragically lead to fatalities [3]. Currently, according to WHO data, COVID-19 has caused approximately 7 million deaths worldwide. 

SARS-CoV-2 displays a wide organotropism, as evidenced by autopsy results (e.g., kidney, heart, intestines, liver, and brain); however, it exhibits a particular preference for infecting the respiratory system [4,5]. Structures facilitating virus entry include angiotensin-converting enzyme 2 (ACE2) and transmembrane serine protease 2 (TMPRSS2), found in the respiratory tract, cornea, and gastrointestinal cells [6]. 

Most infected individuals will have mild respiratory symptoms like cough, with or without sputum, in addition to fever, fatigue, and muscle pain. However, a small percentage of them will progress to develop a hyperinflammatory state with pulmonary edema and cellular infiltration, leading to acute respiratory distress syndrome (ARDS) [7]. The hyperinflammatory state can impact various body systems due to an unregulated innate immune response from the host, resulting in elevated levels of proinflammatory cytokines such as IL-1, IL-6, tumor necrosis factor-alpha (TNF-α), and other acute-phase markers like C-reactive protein (CRP), D-dimer, or ferritin [8]. This “cytokine storm” plays a pivotal role in the development of severe COVID-19, including pulmonary damage and microvascular thrombosis [9]. Additionally, it is responsible for the emergence of complications at the renal level, coagulation disorders, ARDS, or shock, which can lead to mortality rates exceeding 30% [10]. Ongoing studies are investigating various aspects of the immune response to the virus [11]. 

Roughly 10% of severe patients need medical care in intensive care units (ICUs) [12]. In COVID-19, risk factors associated with the progression and severity of the infection have been identified, such as advanced age and various comorbidities, as well as alterations in various laboratory parameters. Among the latter, elevated values in lactate dehydrogenase (LDH); CRP; procalcitonin; ferritin; and proinflammatory cytokines such IL-6, IL-2, IL-1β, TNF-α, and granulocyte colony-stimulating factor (G-CSF) stand out [13,14]. Increased IL-6 levels and hyperferritinemia are regarded as markers of systemic inflammation and an unfavorable prognosis in COVID-19 [15]. IL-6 plays a crucial role in the development of SARS-CoV-2, functioning not only as a proinflammatory cytokine, but also influencing the initiation of coagulation, promoting platelet and leukocyte adhesion. Additionally, persistently elevated IL-6 levels are responsible for the pulmonary fibrotic complications seen in affected patients [16]. The neutrophil-to-lymphocyte ratio is often elevated in patients with severe COVID-19 [17]. 

Besides inflammation, coagulation issues significantly impact infection prognosis. Hypercoagulability raises the risk of thrombotic and hemorrhagic events, whereas the procoagulant state depletes platelets and coagulation factors, resulting in hemorrhagic conditions.

Several studies have found that increasing prophylactic doses of low molecular weight heparin (LMWH) does not appear to confer a benefit, but rather poses an elevated risk of bleeding in patients with severe COVID-19 [18]. Furthermore, the therapeutic use of enoxaparin improves gas exchange and reduces the need for mechanical ventilation (MV) in these cases, unlike prophylactic anticoagulation. 

In another open-label, adaptive, multi-platform, randomized clinical trial that included 1098 patients with severe COVID-19, therapeutic anticoagulation with heparin did not result in a higher probability of survival upon hospital discharge or a greater number of days without cardiovascular or respiratory support compared to standard pharmacological thromboprophylaxis [19]. On the contrary, in non-critical COVID-19 patients, an initial strategy of therapeutic-dose heparin anticoagulation increases the probability of survival upon hospital discharge with reduced use of organ support [20]. 

In patients with non-severe COVID-19 receiving therapeutic-dose heparin, but who eventually develop a severe clinical condition, the continuation of the therapeutic dose compared to dose reduction does not confer any clinical benefit and appears to be harmful [21]. Therefore, the use of prophylactic or therapeutic anticoagulant doses should be individualized based on the patient’s clinical condition and their risk of thromboembolic events.

For a thorough evaluation and continuous monitoring of the coagulation profile in COVID-19 patients, it is imperative to assess an array of laboratory parameters, including prothrombin time (PT) and its activity (PT-act), activated partial thromboplastin time (aPTT), platelet count, antithrombin (AT), D-dimer, and fibrin/fibrinogen degradation products (FDP). Additionally, the PaFi index, representing the ratio between arterial oxygen pressure and the fraction of inspired oxygen (PaO_2_/FiO_2_), plays a crucial role in this comprehensive assessment [22].

Although only a few drugs benefit severe COVID patients, early diagnosis and initial therapy, along with nutritional and organ support, can lead to favorable results. 

Currently, widespread vaccination of the population is the most effective public health measure in the fight against SARS-CoV-2.

Our study’s objective is to utilize machine learning (ML) models to classify severe COVID-19 patients at a higher risk of mortality. ML, as part of artificial intelligence (AI), employs statistical and mathematical algorithms that enable the extraction of patterns from variable data, assisting in making complex decisions [23]. These models are designed to make accurate predictions using data from a multitude of variables, unlike classical statistical models created for making inferences about relationships between variables.

AI tools have been implemented in the fight against COVID-19 in various areas, including drug and vaccine discovery or repurposing [24,25]. 

To the best of our knowledge, this is the inaugural multicenter study that both develops and validates six ML models for forecasting factors linked to an elevated risk of mortality in critically ill patients with SARS-CoV-2 infection throughout the six pandemic waves within the Spanish population.

## 2. Materials and Methods

### 2.1. Data Source

Patient information was sourced from several internal outlets within the hospitals: (1) the electronic medical record (EMR) system, equipped with modules for recording clinical analysis outcomes and electronic prescription of medications; and (2) the ICU electronic prescription program. With this data, a personalized data collection questionnaire (DCQ) was tailored for each patient.

### 2.2. Study Design and Population

This is an observational, longitudinal, retrospective, multicenter study conducted in two high-complexity Spanish hospitals. The General University Hospital of Valencia (GUHV) is a public healthcare facility boasting a capacity of more than 500 inpatient beds. In the year 2021, it efficiently managed 129,856 hospital emergencies. Meanwhile, the Virgen de las Nieves University Hospital of Granada (VNUH) is another public hospital with a substantial capacity, offering over 800 inpatient beds. In the same year, it provided care for 234,259 emergencies. Both institutions are classified as tertiary hospitals and are well equipped with more than 40 ICU beds for patient care. A total of 684 patients hospitalized with severe COVID-19 were included in the study (52.2% male). SARS-CoV-2 infection was confirmed through reverse transcription polymerase chain reaction (RT-PCR) testing of a nasopharyngeal swab between 1 April 2020, and 31 March 2023. Severe COVID-19 was defined as cases where patients received respiratory and/or cardiovascular organ support (high-flow nasal cannula oxygen, invasive or non-invasive mechanical ventilation, extracorporeal life support, vasopressors, or inotropes) in an ICU.

The inclusion criteria were: patients > 18 years old admitted to the GUHV and the UHVN with severe COVID-19 pneumonia, comorbidities, and requiring ICU admission.

The exclusion criteria were: patients ≤ 18 years old, patients with missing data for more than one clinical/analytical variable during this study period.

The participants provided informed consent before participating in the study, and it was approved by the Ethics Committee of GUHV and VNUH.

### 2.3. Study Data

The DCQ included information on demographic, clinical, and laboratory variables, as well as the date of hospital admission. Data on specific pharmacological treatments that could influence the final outcome were also included. For all patients, laboratory parameter data were collected at the time of ICU admission.

The questionnaire was divided into five sections:

(1) Patient characteristics: this section included sociodemographic variables such as age and gender, as well as height and weight, which were used to calculate the body mass index (BMI) in kg/m^2^. This allowed for stratification between patients with normal weight (BMI = 18.5–24.9) and those with excess weight (BMI ≥ 25). The latter was further classified as: (a) overweight (BMI = 25–29.9); (b) grade I or moderate obesity (BMI = 30–34.9); (c) grade II or severe obesity (BMI = 35–39.9); and (d) grade III or morbid obesity (BMI ≥ 40).

Among the clinical variables, the presence of relevant comorbidities was included (hypertension, diabetes mellitus (DM), dyslipidemia, chronic obstructive pulmonary disease (COPD), asthma, obstructive sleep apnea (OSA), use of oxygen therapy (differentiating between invasive and non-invasive), ischemic heart disease, chronic kidney disease (CKD), dementia, smoking, active neoplasia, autoimmune disease, and coagulopathy). Hypertension is defined as a sustained elevation of blood pressure, with office systolic blood pressure values ≥ 140 and/or diastolic blood pressure values ≥ 90 mmHg [26].

If the patient had any other serious underlying pathology, it was specified in an open-text section.

The following symptoms were taken into account at the time of admission: fever, cough, dyspnea, nausea and vomiting, diarrhea, and neurological symptoms.

(2) Pharmacological treatment: previous and/or during hospitalization, the following treatments were taken into account: angiotensin-converting enzyme inhibitors (ACEIs)/angiotensin receptor blockers (ARBs); use of antivirals: oral lopinavir/ritonavir, oral darunavir/cobicistat, or intravenous remdesivir (200 mg on day 1, followed by 100 mg/day for up to 10 days); use of oral hydroxychloroquine; use of subcutaneous interferon beta-1b; use of subcutaneous anakinra, preferably in patients with poor clinical progress (PaO_2_/FiO_2_ < 300, SpO_2_ < 92% in room air, tachypnea, or elevated ferritin levels); dosage of 200 mg/24 h (day 1) and 100 mg/24 h (days 2–5); use of tocilizumab and/or baricitinib or other immunosuppressants such as cyclosporine and corticosteroids; use of N-acetylcysteine; use of antibiotics (e.g., azithromycin); as well as the use of prophylactic or therapeutic doses of anticoagulants.

(3) Analytical data: the closest analysis following hospital admission was taken into account, as well as the first analysis since admission to the ICU. The latest analyses during the hospital stay were recorded. The laboratory parameters collected included hemoglobin, platelets, lymphocytes, albumin, creatine kinase (CK), LDH, CRP, procalcitonin, ferritin, glutamate-pyruvate transaminase (GPT), glutamate-oxaloacetate transaminase (GOT), creatinine, fibrinogen, D-dimer, PT, and aPTT. Additionally, the PaFi index was considered as a gasometric parameter, as it provides information on gas exchange and the possible presence of acute lung injury or even acute respiratory distress syndrome (ARDS).

(4) Procedures performed during hospitalization: the following procedures were included: conventional oxygen therapy, non-invasive ventilation (NIV), invasive mechanical ventilation (IMV), prone positioning ventilation, hemodialysis/hemofiltration, and extracorporeal membrane oxygenation (ECMO).

(5) Patient final outcome: the clinical complications during ICU admission were collected, including acute kidney and/or liver injury with functional failure, acute lung injury with respiratory failure, ARDS, sepsis, and septic shock. Improvement in the patient’s symptoms was assessed considering laboratory tests (including gasometry) and radiological evaluations at 7 and 21 days from admission, depending on the duration of their stay. The number of days of hospitalization until discharge was recorded, either due to clinical improvement or unfortunate outcome.

The final outcome measures taken into account were overall mortality, as well as mortality at 7 and 21 days from admission.

### 2.4. Method

#### 2.4.1. Model Development

In this study, the method based on the extreme gradient boost (XGB) was applied as a reference method. In addition, a comparison was made with other ML systems. It is a flexible, efficient, and portable supervised learning algorithm. The main advantages are its high execution speed, its scalability, it allows parallel computing, and it usually outperforms other algorithms in accuracy in solving many data science problems [27,28,29]. For these reasons, XGB was used in the present study to classify patients with severe COVID-19 and predict variables associated with increased mortality.

Given a dataset *S = x_j_*, *y_j_*, the XGB model was designed using the following:(1)$$\hat{y_{j}}=\sum_{p=1}^{P}t_{p}\left(x_{j}\right)$$
where *x_j_* represents the input vector with *m* time variables, $\hat{y_{j}}$ shows the predicted output, *y_j_* represents the output, *t_p_* shows a tree with leaf weight *w_p_* and structure *u_p_*, j = 1; 2; …; *n*, and *P* corresponds to the number of trees.

The regularized objective function for the proposed method is presented in Equation (2). A second-order Taylor expansion is implemented to approximate the XGB objective function in order to improve the prediction accuracy [30].
(2)$$R=\sum_{j}r\left(\hat{y_{j}},y_{j}\right)+\sum_{p}\Psi \left(t_{p}\right)$$
(3)$$\Psi \left(t_{p}\right)=\lambda f_{p}+\frac{1}{2}\gamma {\left\|\omega_{p}\right\|}^{2}$$

In Equation (3), *f_p_* shows the number of leaves on the tree. The R () function penalizes the complexity of the method. The learning rate is shown by *λ*, and *w_p_* is the vector of leaf scores. To control the complexity weight of the system, a parameter *γ* is employed. The objective is to optimize Equation (2) [28].

In this work, other ML algorithms have been implemented to test the performance of the proposed method. All of them are widely used in the scientific community. The five that gave the best results in the comparison were selected. The following methods are of note: decision tree (DT) [31], Gaussian Naïve Bayes (GNB) [32], Bayesian linear discriminant analysis (BLDA) [33], k-nearest neighbors (KNN) [34], and support vector machine (SVM) [35]. The MatLab Statistical and Machine Learning Toolbox (MatLab 2022a; The MathWorks, Natick, MA, USA) was used to design the models. The database is separated into two blocks; 70% of them are used for training and the other 30% for testing, and patients were not shared. For the validation of the results, a 5-fold cross-validation was performed to avoid overfitting. To reach the optimal point of operation of the ML algorithms, the different hyperparameters of each method are usually adjusted in the training phase. In this study, Bayesian techniques have been used for these hyperparameter values. Bayesian optimization belongs to a class of sequential model-based optimization algorithms that allow us to use the results of our previous iteration to improve our testing method of the next experiment. This, in turn, limits the number of times a model needs to be tested for validation, since only those hyperparameters that are expected to generate a higher validation score are passed for evaluation. With this optimization method, the result of the developed methods is improved. The most prominent hyperparameters of the implemented systems are as follows. To optimize the performance of the XGB system, the hyperparameters eta = 0.15, minimum chil weight = 1, gamma = 0.25, alpha = 0.5, maximum depth = 8, lambda = 0.25, col sample by tree = 0.6, and maximum delta step = 4 have been adjusted. For the SVM method, a Gaussian kernel function is chosen with the following parameters: C = 1.0, sigma = 0.5, numerical tolerance = 0.001, and iteration limit = 200. For the DT system, the base parameter estimator is adjusted: tree, maximum number of divisions = 16, learning rate = 0.1, and number of learners = 60. As for the BLDA algorithm, the Bayesian kernel has been selected. Finally, for the KNN method, the distance metric is Euclidean and it uses 30 neighbors.

In all simulations, 100 iterations were performed to obtain the mean and standard deviation values in a uniformly random manner. In this way, the impact of noise is reduced, the appropriate values are calculated, and statistically valid results are obtained [36]. The phases applied in this study are described in Figure 1. As can be seen, the subjects to be studied were first chosen. Once the database was implemented, training and validation of the ML methods were carried out.

#### 2.4.2. Performance Evaluation

In this work, the different methods were compared with the following metrics: specificity, precision (also known as positive predictive value), recall (also known as sensitivity), balanced accuracy, degenerate Youden index (DYI), F_1_ score, Matthew’s correlation coefficient (MCC), Cohen’s Kappa index (CKI), receiver operating characteristic (ROC), and area under the curve (AUC) [35]. The F_1_ score is described as: (4)$$F_{1}\;score=2\frac{Precision\cdot Recall}{Precision+Recall}$$

MCC was also used to test the performance of the ML methods, defined as:(5)$$MCC=\frac{TP\cdot TN-FP\cdot FN}{\sqrt{\left(TP+FP\right)\left(TP+FN\right)\left(TN+FP\right)\left(TN+FN\right)}}$$
where TP shows the number of true positives, FP represents the number of false positives, TN is the number of true negatives, and FN corresponds to the number of false negatives. CKI was used to estimate the overall performance of the system [37].

## 3. Results

This section describes the results obtained after applying the different ML methods on the data from the records of the different patients with severe COVID-19 to define the factors with the greatest influence on in-hospital mortality. The performance of the proposed method, XGB, has been compared with different ML classification methods accepted in the scientific community.

Table 1 displays the results obtained through various classification methods, including DT, GNB, BLDA, KNN, SVM, and our proposed system for classifying mortality in severe COVID-19 patients. Notably, GNB- and BLDA-based methods exhibit a comparatively lower balanced accuracy, failing to reach the 85% threshold. In contrast, DT and SVM methods demonstrate enhanced classification capabilities, approaching a balanced accuracy of 90%, surpassing the performance of GNB and BLDA.

Conversely, the KNN-based method and our proposed XGB system achieve balanced accuracy scores exceeding 90%, representing a significant improvement over prior methods, resulting in enhanced predictive capabilities. Notably, XGB attains a score exceeding 96%. 

KNN and SVM shine as the algorithms that closely match XGB in terms of precision and recall values, outperforming DT and, notably, surpassing GNB and BLDA in terms of results. Moreover, this pattern is reflected in Table 1 when considering the F1 score parameter, where XGB attains elevated values, serving as a clear indicator of enhanced classification performance.

To evaluate the performance of the proposed XGB system in classifying mortality among patients with severe COVID-19, we computed several widely-used parameters from the literature, including AUC, MCC, DYI, and the kappa index. Among these, the MCC stands out as one of the most reliable statistical indices available, as it yields a high score only when predictions have been accurate across all four categories of the confusion matrix.

The results across these four categories—true positives, false negatives, true negatives, and false positives—are directly influenced by the size of the positive and negative elements within the dataset. As shown in Table 1, the XGB method achieved an MCC value of 86.50%, surpassing the values obtained by KNN (80.79%) and SVM (79.43%). Notably, the other methods displayed inferior performance in this parameter.

Similarly, when examining the kappa index, XGB reached a value close to 87%, representing a substantial improvement over KNN and SVM by 6.61% and 8.19%, respectively. This pattern persists when considering the AUC and DYI parameters, with XGB consistently achieving higher values. These results underscore the superior ability of the XGB method in effectively classifying mortality among patients with severe COVID-19. Figure 2 presents a comprehensive overview of the performance comparison between the XGB method and alternative classifiers across various key metrics, including balanced accuracy, recall, precision, F1 score, CKI, MCC, AUC, and DYI.

Furthermore, the ROC curve serves as a critical tool for assessing and contrasting the classification efficacy of the proposed system against alternative ML methods. This curve is constructed by plotting the sensitivity against specificity across a range of threshold values [36]. Figure 3 visually conveys the outcomes attained by various classification systems in line with the primary goal of patient categorization in this study.

Specifically, the XGB method showcases a substantial area under the ROC curve, emblematic of its enhanced capacity to effectively classify the two distinct classes. This is further corroborated by the numerical values provided in Table 1.

To enhance clarity, we have organized all metrics for each dataset, both training and validation, and visualized them as a radar plot (Figure 4). In an ideal scenario, where the model excels across all metrics, the plot would form a circle encompassing the entire grid. In our study, the training sets consistently demonstrate higher scores across all metrics, whereas the validation sets generally exhibit lower scores.

The shape of these radar plots also offers insights into the model quality. A larger circle area within the validation set signifies a superior predictive method. The proposed XGB system, as depicted in Figure 4, serves as a compelling example of a well-balanced model. Notably, both the training and validation sets yield similar radar plots, underscoring the absence of overfitting or underfitting, thereby enhancing the model’s generalizability. This means that the system performs effectively in delivering accurate outputs when presented with new inputs.

In contrast, the GNB method consistently ranks as the poorest performer across all metrics. Considering these results, we can confidently assert that the proposed XGB system adeptly classifies patients in alignment with the study’s objectives, offering high accuracy and automation—valuable tools for clinical practice.

With the proposed XGB method, the predictor factors or variables associated with a worse outcome in critically ill COVID-19 patients in terms of mortality were: age; high BMI; elevated ferritin, LDH, CRP, and creatinine levels; lymphopenia; low PaFi values; use of IMV; and abnormal values in hemostasis and coagulation, namely elevated PT and aPTT, elevated D-dimer and fibrinogen, and low platelet levels. Figure 5 displays a bar chart illustrating the weights of predictor variables that significantly enhance classification accuracy across various ML methods.

The baseline clinical data of the 684 patients included in the study are shown in Table 2.

Figure 6 provides a graphical representation of the number of selected patients admitted to the hospital with severe COVID-19 throughout the six pandemic waves and until the end of the study.

## 4. Discussion

To our knowledge, this is the first study to develop, compare, and evaluate six supervised ML methods for predicting mortality in severe COVID-19 patients at a Spanish tertiary hospital. We collected data on 66 demographic, clinical, and laboratory variables. After analyzing these algorithms, XGB achieved the highest balanced accuracy at 96.61%.

SARS-CoV-2 invading airway cells leads to cell damage and triggers a local immune response, causing the release of proinflammatory substances such as IFNγ, IL-1β, IL-6, and TNF-α [38]. IL-1β exerts its influence on endothelial and vascular smooth muscle cells, thereby stimulating the production of IL-6 [39]. IL-6 plays a crucial role in the shift from mild inflammation to severe hyperinflammatory conditions, including cytokine release syndrome (CRS), acute respiratory distress syndrome (ARDS), and lung damage, which can lead to high mortality in critically ill COVID-19 patients [40]. ARDS is the leading cause of death among these patients, often associated with mortality rates exceeding 70% [7,41]. ARDS results from an imbalanced immune response, excessive inflammation, and the activation of blood clotting, with CRS at the center of this complex interaction [42]. Current treatment strategies mainly focus on regulating the immune response and achieving antiviral, antithrombotic, or anticoagulant effects [7]. Drugs with substantial evidence for managing CRS include IL-1 inhibitors (e.g., anakinra), IL-6 inhibitors (e.g., tocilizumab), janus kinase inhibitors (JAKi, e.g., baricitinib), and corticosteroids. 

In our study, 33.8% of the patients died in the hospital, and 22.4% of all patients needed invasive mechanical ventilation (IMV). The median hospital stay was 15 days (IQR 9–30.2). Among all the ML classifiers applied, the XGB method was the pattern recognition method that most accurately discriminated between patients at a higher risk of mortality. This model was analyzed and compared with different supervised ML methods described in the literature, such as BLDA, DT, GNB, KNN, or SVM. Current ML classification methods employed in biomedical applications have consistently demonstrated that supervised algorithms, be it for regression or classification, typically achieve higher average accuracy rates compared to their unsupervised counterparts [43]. In our study, GNB and BLDA were the methods with the worst performance, whereas KNN was the method that came closest to the accuracy values of XGB. These results align with findings from studies describing these supervised ML algorithms in predicting COVID-19 mortality [44,45].

Our study uses a radar graph to effectively assess how well ML models perform during training and testing. The results show that the XGB model excels in handling large datasets without overfitting, outperforming other methods with higher precision, recall, and accuracy. This reliability makes XGB a valuable tool in biomedical applications, like predicting cancer patient stages [46].

In our patient cohort, 52.2% were male, and the median age was 63 (IQR 55–74). Consistent with the results from previous studies, advanced age was identified as the primary demographic factor predicting hospital mortality in COVID-19 patients [47,48,49]. Sánchez-Montañes et al. conducted a study that applied various ML methods, and they found age to be the most significant predictor of mortality [48]. Similarly, other authors who utilized different ML models, such as deep learning models [50] or artificial neural networks [51] also emphasized age as a predictive factor for the progression to a severe/critical clinical condition and/or mortality. Furthermore, age stands out as the most significant demographic risk factor for both mortality and ICU hospitalization duration, in addition to being a crucial factor in the days of invasive mechanical ventilation (IMV) among critically ill COVID-19 patients [52]. 

On the other hand, conditions like diabetes, hypertension, heart issues, as well as COPD, asthma, CKD, and cancer, among others, have been recognized as risk factors for disease progression and increased mortality risk in COVID-19 patients using ML techniques [49,53]. In our study, we found that clinical factors related to other health conditions did not have a very strong impact on predicting mortality. However, high levels of creatinine, which indicates kidney problems, stood out as an important predictor of mortality. Additionally, severe cases of COVID-19 are often associated with the development of acute kidney injury, which can ultimately lead to the patient’s death [54,55]. 

Similar to our findings, a recent retrospective study by Datta D et al. confirms, using ML techniques, that elevated BMI is a significant predictor of mortality in these patients [56]. Furthermore, unlike our study, in this research, both diarrhea and smoking status were strongly associated predictors of mortality.

Although age is the primary factor associated with mortality, younger individuals with hypertension, diabetes, and obesity face a similar risk of death as individuals who are 20 years older and do not have any of these three conditions [57]. 

Many studies in Adamidi et al.’s systematic review share some of the predictors for disease progression and mortality that we found in our study. Their review consistently highlighted age, CRP levels, lymphocyte counts, LDH levels, and findings in chest X-rays and CT scans as the most commonly reported indicators linked to a poor outcome in COVID-19 patients [44].

In our investigation, over 20% of the patients exhibited ARDS based on the latest definition [58]. ARDS is a condition of severe oxygen deficiency caused by lung inflammation, not by heart-related lung swelling. Several studies have found that ARDS and associated pneumonia are factors that can lead to mortality in SARS-CoV-2 infected patients [44,49]. For patients like these, using ventilatory support is crucial. Therefore, our study includes the use of IMV as an important predictive factor.

Similar to our research, other studies that employed ML tools have identified various serum biomarkers as prognostic indicators of severity and mortality in patients infected with SARS-CoV-2 [44,47,59,60,61]. Like in our case, predictors that forecasted a higher risk of mortality included, among others, CRP, LDH, and ferritin. As it is well understood, CRP and LDH values tend to rise in the case of severe infection, tissue damage, or injury, as well as in chronic diseases. A systematic review by Bottino et al. concludes, as does our study, that age and CRP and LDH levels are among the predictors most associated with mortality [45].

Elevated levels of proinflammatory cytokines, which are implicated in ARDS, accelerate ferritin synthesis [62]. Consequently, elevated levels of serum ferritin can assist in predicting the development of ARDS and an increased risk of mortality in COVID-19 patients. The meta-analysis conducted by Henry et al. substantiated that serum ferritin, in conjunction with other biomarkers like elevated levels of IL-6 and IL-10, as well as reduced levels of lymphocytes and platelets, were linked to the progression of severe and fatal illness [63]. Serum ferritin is a biochemical parameter that has proven to be a powerful predictor of mortality in severe COVID patients, as highlighted by several studies employing ML tools [64,65]. In fact, in our study, it emerges as the most strongly associated predictive variable with mortality.

SARS-CoV-2 infection can predominantly affect T lymphocytes, specifically CD4+ and CD8+ T cells, leading to a reduction in their counts. Lymphopenia is more commonly observed in severe cases when compared to moderate ones [66]. In a retrospective study by Wang T et al., as in ours, a low percentage of lymphocytes was a predictive variable for mortality in hospitalized COVID-19 patients [60]. In the systematic review by Adamidi ES et al., lymphopenia was also identified as one of the primary predictors of disease progression and mortality in infected patients [44].

On the other hand, the intense inflammatory response triggered by COVID-19 results in significant disturbances in hemostasis and substantial alterations in coagulation parameters [67,68]. The coagulation disorder, resulting from the hyperinflammatory state and altered immune response, has been described as a state of disseminated intravascular coagulation (DIC) and consumption coagulopathy, characterized by a decrease in platelet count, an increase in FDP such as D-dimer, and low fibrinogen levels [68]. These findings may elucidate the venous thromboembolic events observed in some of these patients and provide support for thromboprophylaxis/antithrombotic treatment. Critically ill patients with COVID-19 often present significant coagulation abnormalities, such as thrombocytopenia and widespread arterial and venous thrombosis [69].

In our patient group, we found a high percentage of coagulation disorders, affecting more than 25% of patients. It is well-known that worsening coagulation measurements during the illness are linked to disease progression and a higher risk of death. Moreover, there is a strong connection between abnormal coagulation values and the development of DIC, which is closely tied to the severity and poor outcomes in these patients [22]. 

In contrast to coagulopathy/DIC associated with bacterial sepsis, COVID-19 exhibits a lower incidence of PT and aPTT prolongation, as well as reduced antithrombin activity. Nevertheless, the underlying causes of coagulopathy remain poorly understood. It is hypothesized that factors such as hypoxia, endothelial damage, dysregulated immune responses mediated by inflammatory cytokines, and lymphocyte cell death may contribute to its onset [70].

Despite this, PT remains a fundamental parameter widely utilized in clinical settings to assess coagulation function. 

In line with the findings from Jin et al.’s meta-analysis, our own research revealed a strong correlation between mortality and notably elevated D-dimer levels, extended PT, and decreased platelet counts in comparison to patients who successfully survived [71]. Other studies corroborate that laboratory parameters indicating coagulation abnormalities, such as elevated D-dimer and FDP levels, prolonged PT, and aPTT, are associated with mortality in COVID-19 patients [22,72].

Elevated D-dimer levels and low platelet counts appear to be strong prognostic indicators of coagulation abnormalities in patients with severe COVID-19. These laboratory parameters, along with other coagulation measurements, should be assessed simultaneously with fibrinogen determination to rule out DIC and to monitor antithrombotic treatments in critically ill COVID-19 patients effectively [73]. A low platelet count is linked to an elevated risk of severe illness and mortality in COVID-19 patients, making it a crucial clinical indicator of deteriorating health during hospitalization [74]. Various studies confirm that anticoagulant therapy, primarily with LMWH, is associated with a better prognosis in severe COVID-19 patients with significantly elevated D-dimer levels [75]. 

Thromboelastography is a diagnostic tool for checking excessive blood clotting in COVID-19 patients. It also helps determine if more anticoagulation is necessary. Monitoring coagulation parameters continuously provides a better way to assess how patients are progressing clinically, rather than relying on a single measurement, because blood clotting can change rapidly in critically ill COVID-19 patients [68].

Unlike our study, other studies included prothrombin time activity (PT-act) as a variable, with PT-act < 75% being independently associated with mortality [22].

Numerous research studies have employed a variety of ML techniques, with coagulation parameters playing a pivotal role as mortality predictors in patients infected with SARS-CoV-2. In the majority of these studies, much like in our own work, the most influential predictors of mortality are D-dimer and INR [64,76]. The INR (international normalized ratio) provides a standardized means of interpreting PT results, regardless of the analysis method employed.

In our study, no pharmacological treatment carried enough significance in predicting mortality.

The available ML-based studies on mortality prediction in COVID-19 patients are hindered by the limited sample size, the type of variables used in the prediction, and the short-term follow-up of the study [77,78].

Comparative studies have revealed that ML methods can be more accurate and efficient than traditional logistic regression analysis, especially when the sample size is limited [79].

The XGB method is a binary classification system that is easy to implement and train, which means that as more data becomes available, this algorithm improves its predictive performance [45]. In their systematic review, Sánchez-Salmerón et al. emphasize that the XGB method and deep neural networks are the leading models for accurately predicting critical outcomes, such as mortality or the need for intensive care or hospitalization [80]. 

Just like in our research, other studies also chose to use the XGB method, obtaining similar results in predicting mortality in COVID-19 patients [47,59]. 

This study’s main limitation is its retrospective design; however, this is partially mitigated by the use of powerful methodological tools like ML. To further improve the predictive performance of the model, including data from other sources, such as genomic profiles and medical imaging, could be beneficial.

A notable strength is the inclusion of a diverse patient population with a relatively high prevalence of comorbidities, especially coagulation disorders, which are often underrepresented in clinical trials. This makes our results more applicable to a broader range of patients.

Additionally, because most patients had a hospital stay of more than one week, our model can predict mortality more than a week in advance.

## 5. Conclusions

ML techniques are playing an increasingly crucial role in predicting events of interest, both in general and, specifically, in predicting the severity and mortality of COVID-19. Among the six supervised ML methods analyzed and validated, the XGB method achieved the highest accuracy in predicting hospital mortality in critically ill patients with severe COVID-19. The factors associated with a higher risk of mortality include advanced age; elevated body mass index; the use of IMV; analytical variables such as low PaFi ratio; elevated levels of ferritin, CRP, LDH, and creatinine; lymphopenia; and other parameters associated with hemostasis and coagulation, including low platelet counts, elevated levels of fibrinogen and D-dimer, prolonged PT, and aPTT.

In our study, we observed a significant population of patients with coagulation disorders. The presence of a coagulopathy in these patients suggests the need to consider antithrombotic strategies. Although the optimal antithrombotic strategy has not yet been established, it appears that LMWH at prophylactic or intermediate doses should be considered for these patients after their admission to the ICU or when elevated D-dimer values are observed, reserving therapeutic anticoagulation for cases where clear local or systemic thrombotic pathology is detected. The XGB method has the potential to assist healthcare professionals in making early and effective critical clinical decisions for severely ill COVID-19 patients with a high risk of mortality and in ensuring the efficient allocation of medical resources.

## Figures and Tables

**Figure 1 viruses-15-02184-f001:**
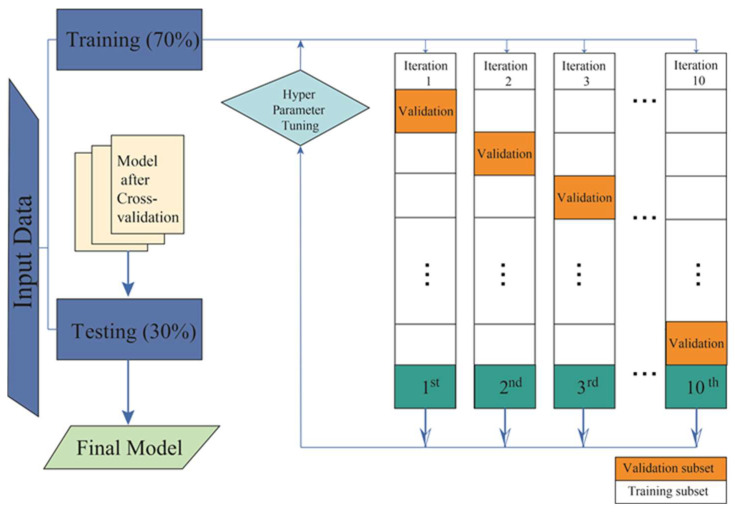
The figure shows the scheme followed in the learning and testing process of this work.

**Figure 2 viruses-15-02184-f002:**
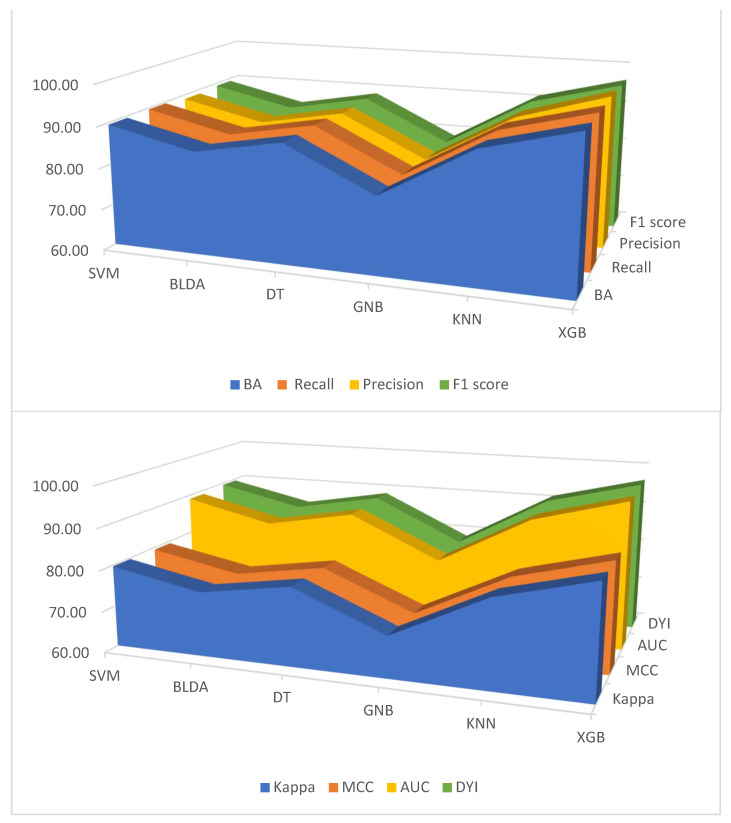
Graphical representation of balanced accuracy (BA), recall, precision, and F1 score values (**top**), and Kappa index, MCC, AUC, and DYI (**bottom**) in percentages. Abbreviations: AUC: area under curve; BLDA: Bayesian linear discriminant analysis; DT: decision tree; DYI: degenerated Younden index; GNB: Gaussian naïve Bayes; KNN: K-nearest neighbors; MCC: Matthew’s correlation coefficient; SVM: support vector machine; XGB: extreme gradient boost.

**Figure 3 viruses-15-02184-f003:**
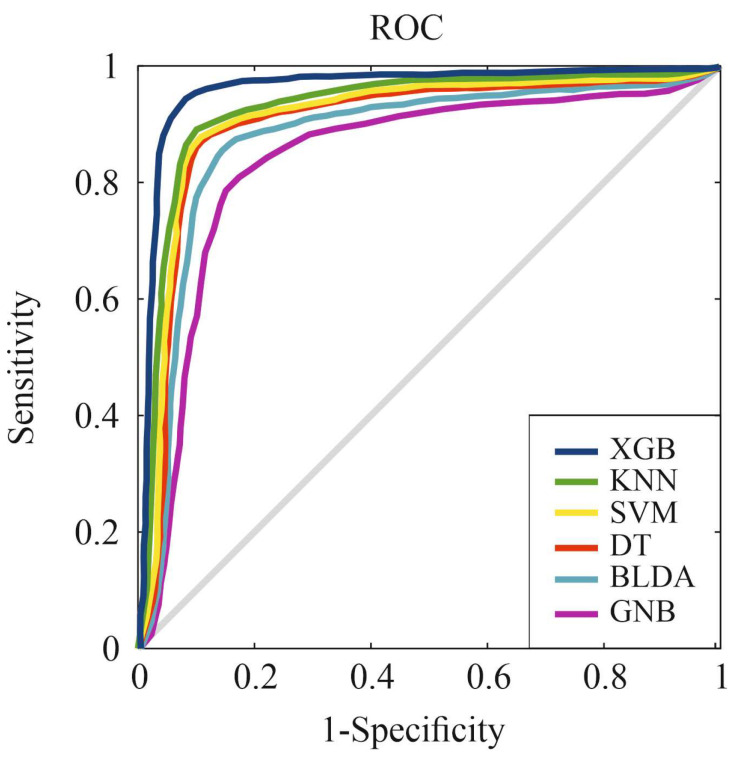
ROC curves for the six assessed machine learning predictors. Abbreviations: BLDA: Bayesian linear discriminant analysis; DT: decision tree; GNB: Gaussian naïve Bayes; KNN: K-nearest neighbors; ROC: receiver operating characteristic; SVM: support vector machine; XGB: extreme gradient boost.

**Figure 4 viruses-15-02184-f004:**
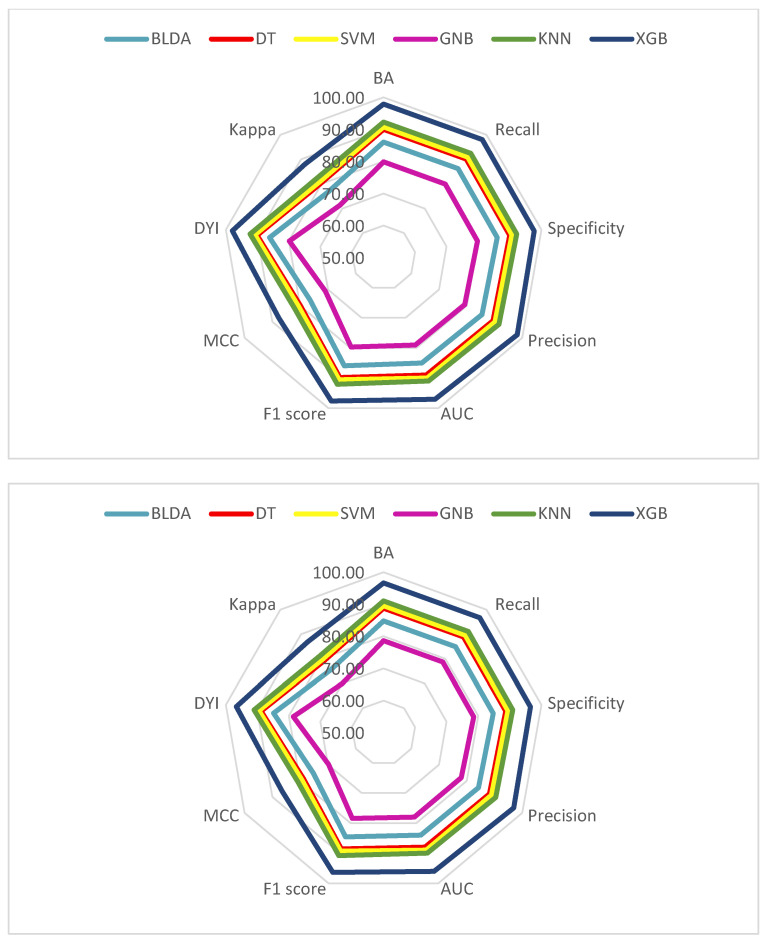
Radar plot of the training phase (**top**) and test (**bottom**) for prediction of mortality in patients with severe COVID-19. AUC: area under curve; BA: balanced accuracy; BLDA: Bayesian linear discriminant analysis; DT: decision tree; DYI: degenerated Younden index; GNB: Gaussian naïve Bayes; KNN: K-nearest neighbors; MCC: Matthew’s correlation coefficient; SVM: support vector machine; XGB: extreme gradient boost.

**Figure 5 viruses-15-02184-f005:**
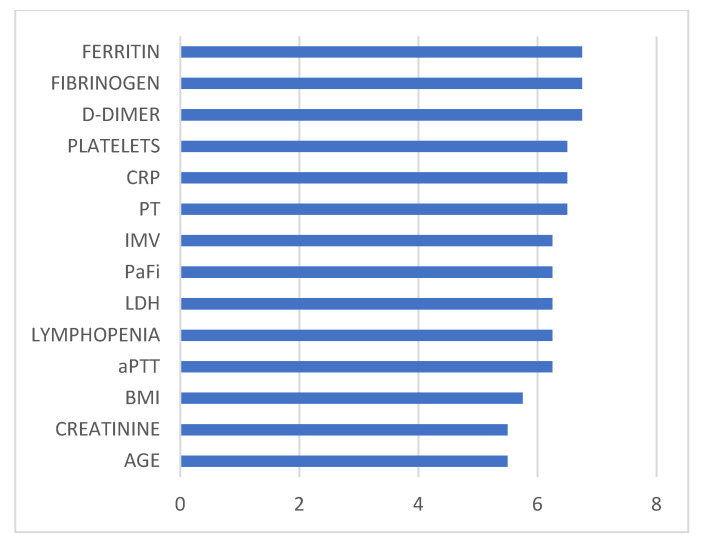
Graphical representation of the predictive variables with the most significant impact on classifying severe COVID-19 patients in terms of mortality. Abbreviations: CRP: C-reactive protein; PT: prothrombin time; IMV: invasive mechanical ventilation; PaFi: ratio between arterial oxygen pressure and the fraction of inspired oxygen (PaO_2_/FiO_2_); LDH: lactate dehydrogenase; aPTT: activated partial thromboplastin time; BMI: body mass index.

**Figure 6 viruses-15-02184-f006:**
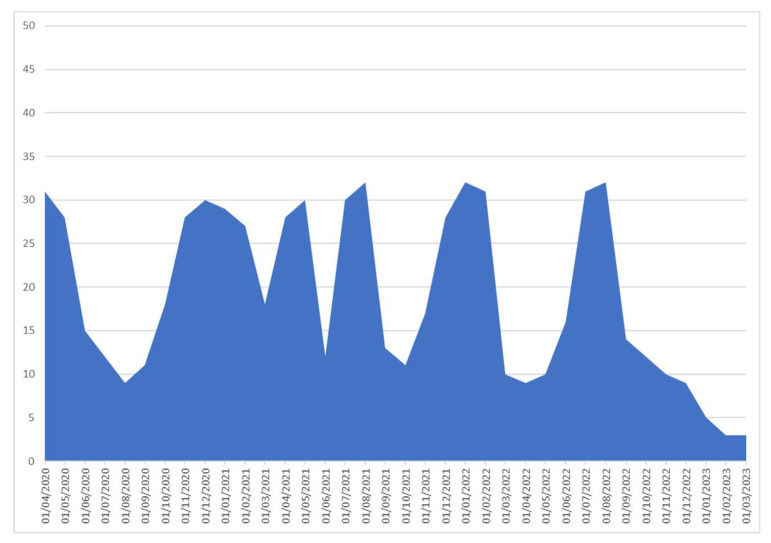
Graphical representation of the number of patients included in the study according to dates throughout the six pandemic waves and until the end of the study.

**Table 1 viruses-15-02184-t001:** Summary of the mean values and standard deviation of balanced accuracy, recall, precision, *F*_1_ score, AUC, MCC, DYI, and Kappa index of the machine learning models and the proposed method implemented in this study.

Methods	Balanced Accuracy	Recall	Precision	*F*_1_ Score
DT	88.78 ± 0.67	88.89 ± 0.71	88.15 ± 0.67	88.52 ± 0.79
GNB	78.61 ± 0.86	78.71 ± 0.80	78.05 ± 0.82	78.38 ± 0.96
BLDA	84.84 ± 0.84	84.94 ± 0.79	84.23 ± 0.86	84.58 ± 0.82
KNN	91.05 ± 0.70	91.16 ± 0.63	90.40 ± 0.66	90.78 ± 0.62
SVM	89.52 ± 0.75	89.63 ± 0.75	88.88 ± 0.80	89.25 ± 0.75
XGB	96.61 ± 0.49	96.74 ± 0.40	96.92 ± 0.48	96.31 ± 0.50
**Methods**	**AUC**	**MCC**	**DYI**	**Kappa**
DT	0.88 ± 0.080	78.78 ± 0.62	88.78 ± 0.78	79.04 ± 0.76
GNB	0.78 ± 0.083	69.75 ± 0.88	78.61 ± 0.94	69.98 ± 0.90
BLDA	0.84 ± 0.074	75.27 ± 0.76	84.84 ± 0.76	75.54 ± 0.80
KNN	0.90 ± 0.067	80.79 ± 0.64	91.05 ± 0.66	81.06 ± 0.62
SVM	0.89 ± 0.087	79.43 ± 0.80	89.52 ± 0.87	79.69 ± 0.76
XGB	0.96 ± 0.048	86.50 ± 0.42	96.61 ± 0.48	86.80 ± 0.49

Abbreviations: AUC: area under curve; BLDA: Bayesian linear discriminant analysis; DT: decision tree; DYI: degenerated Younden index; GNB: Gaussian naïve Bayes; KNN: K-nearest neighbors; MCC: Matthew’s correlation coefficient; SVM: support vector machine; XGB: extreme gradient boost.

**Table 2 viruses-15-02184-t002:** Basal clinical data of patients. Data are *n* (%) or median (IQR) unless otherwise stated.

Variable	Cohort
Number of patients	684
Age (years) (IQR)	63 (55–74)
Male, *n* (yes%)	357 (52.2)
Hospital admission (days) (IQR)	15.0 (9.0–30.2)
Exitus, *n* (yes%)	231 (33.8)
7-day mortality, *n* (yes%)	65 (9.5)
21-day mortality, *n* (yes%)	102 (14.9)
IMV, *n* (yes%)	153 (22.4)
Hypertension, *n* (yes%)	361 (52.8)
Diabetes, *n* (yes%)	183 (26.8)
Dyslipemia, *n* (yes%)	256 (37.4)
Smoker/ex-smoker, *n* (yes%)	156 (22.8)
Asthma, *n* (yes%)	64 (9.4)
COPD, *n* (yes%)	72 (10.5)
OSA, *n* (yes%)	85 (12.4)
BMI, *n* (yes%) - Overweight (BMI= 25–29.9 kg/m^2^), *n* (yes%) - Grade I or moderate obesity (BMI = 30–34.9 kg/m^2^), *n* (yes%) - Grade II or severe obesity (BMI = 35–39.9 kg/m^2^), *n* (yes%) - Grade III or morbid obesity (BMI ≥ 40 kg/m^2^), *n* (yes%)	423 (61.8)
	176 (41.6)
	85 (20.1)
	63 (14.9)
	17 (4.0)
Ischemic heart disease, *n* (yes%)	61 (8.9)
Chronic kidney disease, *n* (yes%)	35 (5.1)
Autoimmune disease, *n* (yes%)	69 (10.1)
Coagulation disorder, *n* (yes%)	179 (26.2)
Active cancer, *n* (yes%)	17 (2.5)
Dementia, *n* (yes%)	65 (9.5)
Fever, *n* (yes%)	412 (60.2)
Cough, *n* (yes%)	485 (70.9)
Dyspnoea, *n* (yes%)	502 (73.4)
Nausea and vomiting, *n* (yes%)	78 (11.4)
Diarrhea, *n* (yes%)	127 (18.6)
Neurological symptoms, *n* (yes%)	56 (8.2)
Angiotensin-converting enzyme inhibitors/angiotensin receptor blockers, *n* (yes%)	325 (47.5)
Antibiotics, *n* (yes%) - Azithromycin, *n* (yes%)	607 (88.7)
	569 (83.2)
Antivirals drugs, *n* (yes%) - Lopinavir/ritonavir, *n* (yes%) - Darunavir/cobicistat, *n* (yes%) - Remdesivir, *n* (yes%)	421 (61.5)
	182 (26.6)
	145 (21.2)
	71 (10.4)
Immunosuppressants and/or immunomodulators, *n* (yes%) - Corticosteroids, *n* (yes%) - Anakinra, *n* (yes%) - Tocilizumab, *n* (yes%) - Baricitinib, *n* (yes%) - Interferon-beta, *n* (yes%) - Cyclosporine, *n* (yes%)	625 (91.4)
	594 (86.8)
	326 (47.6)
	235 (34.3)
	128 (18.7)
	84 (12.3)
	34 (4.9)
Hydroxychloroquine, *n* (yes%)	409 (59.8)
N-acetylcysteine, *n* (yes%)	276 (40.3)
Anticoagulants, *n* (yes%) - Prophylactic heparin, *n* (yes%) - Therapeutic heparin, *n* (yes%)	581 (84.9)
	474 (69.3)
	78 (11.4)
Albumin (g/dL) (IQR)	4.3 (2.8–5.9)
Haemoglobin (g/dL) (IQR)	13.2 (12.4–15.6)
CRP (mg/L) (IQR)	16.5 (12.1–27.6)
LDH (U/L) (IQR)	715.0 (572.0–1034.0)
Procalcitonin (ng/mL) (IQR)	3.7 (0.5–9.8)
Ferritin (µg/L) (IQR)	1195.0 (526.0–1421.0)
Creatinina (mg/dL) (IQR)	1.8 (1.3–2.8)
GPT (U/L) (IQR)	41.0 (27.8–63.8)
GOT (U/L) (IQR)	47.0 (32.6–65.4)
CK (U/L) (IQR)	228.0 (156.0–281.0)
FiO_2_ (%) (IQR)	36.4 (27.3–48.4)
PaFi (IQR)	197.0 (141.0–249.0)
Lymphocytes (10^9^/L) (IQR)	0.6 (0.5–1.3)
Platelets (10^9^/L) (IQR)	153.0 (162.0–315.0)
Fibrinogen (mg/dL) (IQR)	387.0 (304.0–403.0)
D-dimer (ng/mL) (IQR)	1981.0 (1643.0–3256.0)
PT (sec) (IQR)	13.9 (13.2–15.5)
aPTT (sec) (IQR)	34.8 (28.7–37.8)

Abbreviations: IQR: interquartile range, IMV: invasive mechanical ventilation, COPD: chronic obstructive pulmonary disease, OSA: obstructive sleep apnea, BMI: body mass index, CRP: C-reactive protein, LDH: lactate dehydrogenase, GPT: glutamate-pyruvate transaminase, GOT: glutamate-oxaloacetate transaminase, CK: creatine kinase, FiO_2_: inspired oxygen fraction, PaFi: ratio between arterial oxygen pressure and inspired oxygen fraction (PaO_2_/FiO_2_), PT: prothrombin time, aPTT: activated partial thromboplastin time.

## Data Availability

The datasets employed and analyzed in the current study are accessible upon reasonable request from the corresponding author.

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
