# Peer review of "Predictive Model for Mortality in Severe COVID-19 Patients across the Six Pandemic Waves"

_viruses, 2023, doi:10.3390/v15112184_

Round 1
Reviewer 1 Report
Comments and Suggestions for Authors
In the paper entitled "Predictive Model for Mortality in Severe COVID-19 Patients 2 Across the Six Pandemic Waves" the authors present a novel machine learning model based on extreme gradient boosting for the classification of patients COVID-19. The aim of this study is to provide evidence on the most influential predictors of mortality among critically ill patients with severe COVID-19, using machine learning (ML) techniques. To achieve this, they have conducted a retrospective multicenter investigation involving patients with severe COVID-19, from June 1, 2020 to March 31, 2023. It should be noted that they have patients from all different waves of COVID-19. Therefore, the study presents a high significance and obtains relevant results.
• The work is very interesting, the authors must be congratulated.
• The number of patients involved in research is considerable.
• The study is well structured and I believe it can be an interesting contribution to the target audience of this journal.
However, I have some suggestions to address some points in the manuscript.
1. More references should be added in the introduction section.
2. Some studies could be added in the discussion. The conclusion correctly explains the contribution of the work.
3. The description of the method should include an explanation of the hyperparameters used.
4. More information should be added about the methods compared (BLDA, SVM, DT, KNN).
5. Could this type of system be implemented in healthcare systems?
Author Response
Attached file

Reviewer 2 Report
Comments and Suggestions for Authors
The writing in the material and methods must be improved. It says “Patient data were obtained from various internal sources within the hospital,” but later it is stated that there are two hospitals.
The material and methods should be reorganized, starting by saying that the study was conducted in two hospitals. It would be necessary to say what characteristics hospitals have, whether are they public or private hospitals, are they rural hospitals, how many beds they have, what type of hospitals are they, etc.
Explain in one sentence why 70% of the database has been chosen for training and not another percentage (75%, 80%, etc.).
Briefly explain in the discussion if there is any advantage to using Matlab in machine learning over other programs, for example, Python or R? This may interest other researchers.
Perform a comparison of the areas under the ROC Curve. This can be easily done with many programs, for example, Stata.
Please indicate the diagnostic criteria for hypertension in terms of blood pressure.
Consistently use decimals in Table 2. There are times that numbers have a decimal and other times they do not.
In the ethical aspects section, include each hospital's approval date and IRB approval number.
Author Response
Attached file

Reviewer 3 Report
Comments and Suggestions for Authors
I’m sorry but after six waves I find several concerns with this research article:
-not innovative data have been reported after six waves
-after six waves of infection the number of patients of this cohort is low
-I have great attention for machine learning , however I can’t miss to underline several missing clinical points: biomarkers as d-dimer and fibrinogen have been reported to have prognostic role since first wave and related references reported are scarse. Furthermore, during first waves an associated increased rate of pulmonary embolism has been reported but not discussed as relevant comorbodity in severe covid-19 and this is a great limitation.
- diseases associated to patients ‘ frailty with therapeutic support of antiviral or monoclonal antibodies are not shown, maybe an additional table could be useful.
- the chronic comorbidities associated in this epidemiological analysis as diabete, hypertension and cold are also the most common present in general population so they are less importante than acute kidney failure and this fact should be better discussed.
- discussion is too much long and based on data that could be reduced in length vs data that need discussion as reported in my score.
Author Response
Attached file

Round 2
Reviewer 3 Report
Comments and Suggestions for Authors
I carefully red revised version of the manuscript and pbp made by authors and I noted several changes and improvements; yet , reading I confirm that I do not find any new information on the topic.
i’m really sorry to reject it again.